# Assessment of Chilling Requirement and Threshold Temperature of a Low Chill Pear (*Pyrus communis* L.) Germplasm in the Mediterranean Area

**Filippo Ferlito** [1] , **Mario Di Guardo** [2,*] , **Maria Allegra** [1] , **Elisabetta Nicolosi** [2] , **Alberto Continella** [2] , **Stefano La Malfa** [2] , **Alessandra Gentile** [2] and **Gaetano Distefano** [2]

1   CREA, Research Centre for Olive, Fruit and Citrus Crop, Corso Savoia, 190-95024 Acireale (CT), Italy; filippo.ferlito@crea.gov.it (F.F.); maria.allegra@crea.gov.it (M.A.)

2   Department of Agriculture, Food and Environment (Di3A), University of Catania, Via Valdisavoia 5, 95123 Catania, Italy; enicolo@unict.it (E.N.); acontine@unict.it (A.C.); slamalfa@unict.it (S.L.M.); gentilea@unict.it (A.G.); distefag@unict.it (G.D.)

*   Correspondence: mario.diguardo@unict.it

**Abstract:** In temperate climates, bud break and shoot and flower emission of deciduous fruit tree species are regulated by precise chilling and heating requirements. To investigate this aspect, sixty-one accessions of European pear (*Pyrus communis* L.) collected in Sicily were phenotyped for three consecutive years for harvest date, bud sprouting and blooming to determine both the chilling requirements and the threshold temperature using the Chill Days model. The whole germplasm collection was grown in two different experimental fields located at 10 and 850 m above sea level representing two Mediterranean-type climates in which pear is commonly cultivated. Results revealed a mean threshold temperature of 6.70 and 8.10 °C for the two experimental fields, respectively, with a mean chilling requirement ranging from −103 and −120 days. Through this approach, novel insights were gained on the differences in chilling requirement for early flowering cultivars to overcome dormancy. Furthermore, to better dissect differences in chilling requirement between accessions, the sprouting bud rate of six cultivars was assessed on excised twigs stored at 4 ± 0.1 °C from 300 to 900 h followed by a period at 25 ± 0.1 °C varying from seven to twenty-eight days. Results of both experiments highlighted that Sicilian pear germplasm is characterized by a low chilling requirement compared to other pear germplasm, making Sicilian local accessions valuable candidates to be used for selecting novel cultivars, coupling their low chilling requirements with other traits of agronomical interest.

**Keywords:** global warming; bud break; bud sprouting; chill-day model; dormancy

## 1. Introduction

The Mediterranean area is characterized by a wide range of different microclimates, ranging from arid/semiarid to temperate and humid. In the last decades, several global and regional climate models indicated the Mediterranean scenario as a 'Hot-Spot' for future climate change prevision [1,2]. In this region, several areas are experiencing both a general increase in temperatures and a decrease in rainfall. This trend could influence the biological behavior of plants with direct repercussion on their distribution and alter the physiology and the phenology of many species of agronomical interest [3–7].

Temperature plays a fundamental role in promoting bud dormancy and bud break during winter and spring, respectively. In particular, bud dormancy is broken when the plant undergoes a period at low temperatures, while the vegetative and reproductive restart (bud break) is directly influenced by warm temperature. The cold exposure needed to bud break is called a chilling requirement (CR). This stage is followed by the break of quiescence when plants fulfill their heat requirements (HRs) [8–11]. When CR is not completely

fulfilled, the subsequent flowering is characterized by phenological alteration and a lower and irregular flower intensity, with direct repercussion on fruit yield and quality [12–16]. The tight relation between air temperature and vegetative and reproductive development was assessed by several researchers [17–20]. In particular, Chmielewski and Rötzer [21] reported that a 1 °C increase in temperature determined an anticipation of one-week in bud break. Therefore, the set-up of algorithms modeling the influence of temperature on crop phenology can provide useful information that can be readily translated into agronomical practices [22]. Several models were tested in the open field to assess the CR necessary to break dormancy, with the calculation of the Growing Degree Days (GDD) being the most widely known. The Chill Hour (CH) [23], the Utah (CU) [24], the Dynamic (CP) [12], the North Carolina (NC) [25], the Low Chilling (LC) [26] and the Positive Chill Unit (PC) models are others [27]. Other models are based on the calculation of the Growing Degree Hours (GDHs) [24] for the measurement of the HR. Recently, the Chill Days (CD) model [28] has proved its effectiveness on several fruit tree (pear, olive, cherry) and Mediterranean forest species. The CD model is based on the prediction of both the chill days (Cd) needed to overcome dormancy and the antichill days (Ca) for the subsequent bud break.

In some cases, to validate the models, and to better determine the temperature requirements needed for breaking dormancy, additional analyses were performed on excised twigs stored in growth chamber to precisely quantify the number of cold hours needed for bud sprouting [8,29]. This method is based on the quantification of the regeneration of generative organs on branches subjected to different cold temperatures regimes followed by heat treatments to stimulate bud sprouting. This biological test is the only one that allows the determination of chilling requirements, as already demonstrated for several crops such as peach [30], apple [31] and pear [30].

The identification of the most suitable varieties in a given environment (latitude, altitude, slope exposition, etc.) is a key factor affecting the economic success of pear cultivation as well as many other temperate woody species (see [32] for a review on chilling and heat requirements of stone fruits belonging to genus Prunus). The most widely cultivated pear cultivars were selected during the 19th century and are mostly characterized by a high CR (up to 850 CHs > +7 °C), while the availability of cultivars characterized by low CR (particularly priced for their precocity) is limited. Among the species of the genus Pyrus, Japanese pear [*P. pyrifolia* (Burm.f.) Nakai], cultivation has recently shown a sharp increase in warm-winter countries such as Mexico, South Africa, New Zealand and Brazil, even if, in some areas of these countries, the lack of winter chilling hampered bud break. To overcome this problem, a breeding program based on the hybridization between European pears (*P. communis* both high fruit quality and high CR in winter) and Eastern pears (*P. pyrifolia* lower fruit quality and low CR), has generated cultivars combining optimal fruit quality with low CR (thus adapted to subtropical climate) [33].

The development of novel cultivars characterized by low CR could be of great interest in the Mediterranean basin as well. To this extent, an ex situ pear collection encompassing local varieties, national and international cultivars was established and genetically characterized [34,35]. Sicilian pear germplasm displays a high genetic and phenotypic variability for many traits of agronomical interest such as fruit size, flowering and ripening periods, adaptability to limiting environmental factors and resistance to biotic stresses [34]. Among these, the early flowering cultivars could represent valuable genetic sources for breeding programs aimed at obtaining novel cultivars that can be cultivated at lower latitude and altitude.

The objectives of this paper are to: (1) evaluate the phenological behavior of 61 European pear accessions grown in the open field in two Mediterranean environments characterized by marked differences in terms of pedoclimatic conditions; (2) detect the optimal threshold temperature and chilling requirement for the 61 accessions by analysis using the CD model; (3) assess the chill hours needed to break endodormancy in excised twigs for a subset of selected accessions.

## 2. Materials and Methods

### 2.1. Plant Materials, Site Description and Experimental Design

This research was conducted on three consecutive growing seasons from 1 October 2014 to 30 June 2017, in two experimental fields (EF) located in Catania district (Sicily, South Italy), the experimental farm of Catania University (EF1, 10 m a.s.l.) and the Germplasm Bank of 'Parco dell'Etna' located in the field of the Etna volcano (EF2, 850 m a.s.l.). Trees were grafted on the same rootstock and planted in the same year in both EFs. Accessions held in both EFs were clones of the same source tree. In Table 1, for each experimental field, the geographical characteristics, the elevation, the reached Chilling hours (CHs) (Weinberger, 1950) and Chill Units (Richardson et al., 1974), calculated from 1 October to 28 February from 2014 to 2017, are reported. Moreover, the growing degree hours (GDHs) according to Richardson et al. (1974), calculated as the number of hours accumulated between the end of dormancy and the end of fruit set from 1 March to 30 June from 2014 to 2017, are reported.

**Table 1.** Geolocations and yearly and average data related to the Chill Hours, Chill Units (from 1 October to 28 February) and Growing Degree Hours (from 1 March to 30 June) (±standard deviation) calculated from data of the years 2014–2017 in the experimental field 1 (10 m above sea level) and experimental field 2 (850 m above sea level). Climatic data were provided by the Sicilian Water Observatory (www.osservatorioacque.it (accessed on April 2019).

|  | Latitude | Longitude | Elevation [m] | Year | Chill Hours (Hours < 7 °C) | Chill Unit | Growing Degree Hours |
|---|---|---|---|---|---|---|---|
| Experimental field 1 | 37°24′32.52″ | 15°03′16.95″ | 10 | 2014–2015 | 365 | 860 | 1.135 |
|  |  |  |  | 2015–2016 | 345 | 778 | 1.105 |
|  |  |  |  | 2016–2017 | 387 | 987 | 984 |
|  |  |  |  | Average | 366 ± 21 | 875 ± 105 | 1075 ± 80 |
| Experimental field 2 | 37°37′55.40″ | 15°01′17.90″ | 850 | 2014–2015 | 1.237 | 1.934 | 690 |
|  |  |  |  | 2015–2016 | 1.052 | 1.913 | 587 |
|  |  |  |  | 2016–2017 | 1.298 | 2.002 | 563 |
|  |  |  |  | Average | 1196 ± 128 | 1950 ± 47 | 613 ± 67 |

In Figure 1, for each EF, the daily maximum, mean and minimum air temperature and rainfall, registered over a period of 30 years (1984–2013) are reported. Both EFs were established in 2007, with plants grafted onto pear seedlings and subjected to standard agronomical practices. The germplasm consisted of 61 cultivars (each of those was planted in triplicates) as already described by Bennici and colleagues [34]; 18 of those were cultivated in both EFs (Supplementary Table S1).

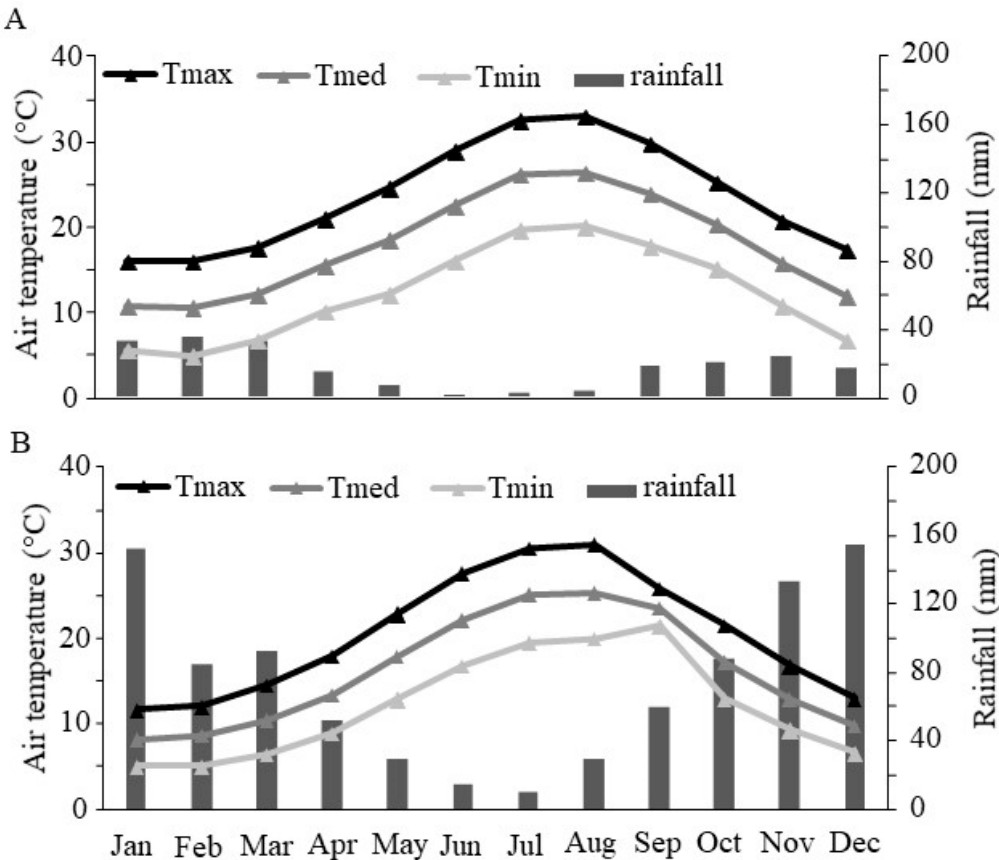

**Figure 1.** Monthly minimum, mean and maximum air temperatures and rainfall registered in the experimental field 1 (10 m a.s.l) (**A**) and experimental field 2 (850 m a.s.l.) (**B**). Data were calculated over a 30 year period from 1985 to 2014. Daily data over the same period are provided in Supplementary Table S2. Climatic data were provided by the Sicilian Water Observatory (www.osservatorioacque.it (accessed on April 2019)).

### 2.2. Phenological Monitoring in Open Field

The main phenological growth stage was monitored every four days according to the Biologische Bundesanstalt, Bundessortenamt and Chemical industry—BBCH [36] from the rest period till the end of blooming. Moreover, for each cultivar, the start of harvest was registered.

### 2.3. Chill Day Model Employment

The Chill Day model (CD) [28] was tested in both EFs. The threshold temperature (TC) and the chilling requirement (CR) values were detected through an iterative process aimed to identify the TC-CR combination that minimizes the root mean square error (RI) between predicted and observed number of days from the end of the previous season (harvest) to bud burst. Analysis was conducted using an in-house R script [37] available upon request (Figure 2). In particular, the CD model accuracy was tested with the root mean square error (RMSE) between predicted and observed dates:

$$\text{Rmse} = \frac{\sqrt{\sum_{i=1}^{n}(dpi - doi)^2}}{N}$$

where *dpi* is the predicted and *doi* is the observed bud-burst date for the ith season, and *N* is the number of seasons.

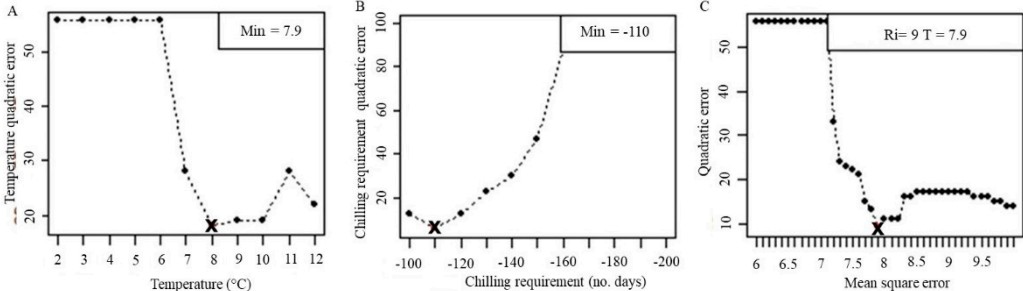

**Figure 2.** Output of the R script employed for the definition of the best values for the threshold temperature (TC) (**A**) and the chilling requirement (CR) (**B**) found by the iteration that minimizes the root mean square error (RI) (**C**), between predicted and observed number of days from the end of the previous season (harvest) to bud-burst (figures referred to 'Coscia' cultivar).

### 2.4. Chill Requirements Analysis in Growth Chamber

A total of six cultivars were selected for the assessment of the chilling requirement after storage in a climatic chamber ('Bianchetto', 'Coscia', 'Gentile', 'Muscatello', 'Ucciardone', 'Urzì'). For each cultivar, 40 twigs of about 30 cm were randomly collected in autumn. The number of nodes per twig varied according to the cultivar architecture from a minimum of four to a maximum of 16. Residual leaves were hand-thinned, and the apical bud was excised to facilitate the switch from para-dormancy to endo-dormancy [38]. The axillary buds from a typical node with one central leaf bud and two flower buds were conserved. Cuttings were bundled into four groups, each containing 10 twigs, and wrapped in paper. Bundles were submerged in water and sterilized with benomyl [methyl-(butylcarbamoyl)-2-benzimidazolecarbamate], rinsed for 10 min, placed in sealed plastic bags and stored at $4 \pm 0.1\ °C$ for 300, 500, 700, 900 h, respectively [8]. After each chilling interval, the twigs were placed with their basal tip in water and forced in a growth chamber at $25 \pm 0.1\ °C$ for 28 days with a photoperiod of 16 h of light. Flower bud-break was assessed at 7 (T7), 14 (T14), 21 (T21) and 28 (T28) days. The bud-break was considered reached when the inflorescence emergence stage (Principal growth stage 5: inflorescence emergence. BBCH stage 53- Bud-burst: green leaf tips and visible flowers) was reached [36].

The end of the endo-dormancy was determined when the bud break exceeded 70% and no further increase was observed [39].

### 2.5. Statistical Analysis

Data related either to chilling requirements, threshold temperature and twigs were analyzed with IBM SPSS Statistics for Windows, version XXI (IBM Corp., Armonk, NY, USA) for analysis of variance (One-way ANOVA) by testing the significance of each variable. Fisher's LSD pairwise comparison procedure at $p \leq 0.05$ level was used to determine significant differences among inflorescence emergence rates observed in excised twigs of six accessions cultivated in the experimental field 2 (850 m a.s.l.) (Supplementary Table S3).

## 3. Results and Discussion

In this research, 61 accessions from Sicilian pear germplasm were grown in two different pedoclimatic areas representatives of different Mediterranean environments in which *P. communis* L. is usually cultivated (Supplementary Table S1) [40,41]. In EF1, monthly maximum temperatures range between 16 °C and 31 °C, while monthly minimum temperature spans from 11 °C to 25 °C in winter and summer, respectively. The total annual rainfall, based on long-term observations, is 208 mm. Tree fruit crops represent the main agricultural activity that, during the centuries, has been set up in the different environments of the Etna volcano. In fact, thanks to the wide varieties of microclimates that are present at different heights and sides of the volcano, it is possible to cultivate many

different species and with significant differences in the harvesting period as well. Wine grape, citrus and olive are the widespread species, but of peculiar importance are pistachio, nut and prickle pear apple, cherry, hazelnut and chestnut. Such cultivation strongly marks the landscape, the economy, and the culture. Although old genotypes are found on all slopes of the volcano, nowadays pears are largely cultivated in the west side of the volcano. In EF2 a particular climate is characterized by cold and wet winters and mild summers in which rain can occur. In the whole year, precipitation exceeds 900 mm. The mean lowest temperature in winter is 6 °C while in summer the monthly maximum temperature is 30 °C (Figure 1, Supplementary Table S2). This area is strictly related to citrus cultivation. In the last few years, some fruit tree crop species such as peach and apricot cultivars with a low chilling requirement, have been cultivated.

The analyzed accessions were previously characterized both morphologically [42,43] and genetically [34,35] and represent a valuable germplasm source for several traits of agronomical interest, such as ripening and harvesting periods, yield and adaptability with limiting environmental factors, and resistance to biotic stress both for fresh and transformed products, or for their antioxidant power [44,45] An assessment of their cold and warm needs to ensure optimal phenological development was carried out in both EF1 and EF2. The analysis was aimed to better dissect the genotype x environment interaction and to provide useful information to growers and breeders for a rational selection of the cultivar(s) for specific environments. Furthermore, the contemporary analysis of the germplasm in two environments is of great interest for a consistent and predictable investigation of the winter chilling accumulation. This aspect is of particular interest, especially in light of the great variability observed between the years of the observations [46]. The annual harvest dates of the two EFs were assessed and implemented in the CD model [28] to calculate the chilling days (Cd), the antichill days (Ca) and the corresponding chilling requirements and threshold temperature for each cultivar in each EF (Tables 2 and 3). These parameters allowed a precise estimation of the days needed to break the bud eco-dormancy and endo-dormancy, respectively. An accurate comparison of the performances and reliability of the most widely used models was carried out by Cesaraccio et al. [28]. Among the tested models, the CD model showed a much lower RMSE compared to the others due to a more accurate determination of the dormancy period.

Significant differences between the two EFs were registered in terms of mean Tc and CR (Table 4, Figures 3 and 4). In the EF1, a lower amount of CR (−103 days) and a higher mean TC (8.1 °C) compared to EF2 (Chilling requirements = −122 days; Threshold Temperature = 6.7 °C) were needed for bud break (Table 4). The observed differences in TC between the two EFs can be due, to a certain extent, to differences in the genetic background between the two germplasm collections, but certainly environment played a significant role. In EF1, the mean RMSE was 2.82 and ranged between 1.94 and 3.17. In EF2 values ranged between 1.94 and 3.08.

**Table 2.** Main phenological stages employed in the Chill Day model for the pears cultivated at the experimental field 1 (10 m a.s.l.). Rows were ordered according to the bud sprouting data. For each accession, harvest date, Chill Days, chilling requirement, antichill days, bud sprouting date, threshold temperatures and root mean square error are reported. Chill days, threshold temperatures and root mean square error are presented both for the three years tested and as average values. Harvest date refers to the previous year compared to the sprouting bud. Accessions in bold were planted in both experimental fields. Chilling requirement and bud sprouting are expressed as doy (day of the year).

| Cultivar | Harvest-Day of the Year | Chill Days (Cd) (No. of Days) | | | | Chilling Requirement ($C_R$) (doy) | Anti-Chill Days (Ca) (No. Days) | Bud Sprouting (doy) | Threshold Temperature ($T_C$) (°C) | | | | Root Mean Square Error (RMSE) | | | |
|---|---|---|---|---|---|---|---|---|---|---|---|---|---|---|---|---|
| | | 2014–2015 | 2015–2016 | 2016–2017 | Average | | | | 2014–2015 | 2015–2016 | 2016–2017 | Average | 2014–2015 | 2015–2016 | 2016–2017 | Average |
| Azzone | 227 | −100 | −119 | −100 | −106 | 333 | 103 | 71 | 11 | 8 | 7 | 8.7 | 3.32 | 2.86 | 2.65 | 2.94 |
| **Gentile** | 186 | −100 | −100 | −110 | −103 | 289 | 148 | 72 | 11 | 7 | 7 | 8.3 | 4.66 | 2.76 | 3.53 | 3.65 |
| **Pasqualino** | 275 | −100 | −100 | −110 | −103 | 13 | 59 | 73 | 11 | 7 | 7 | 8.3 | 4.98 | 4.08 | 4.56 | 4.54 |
| **Virgolese** | 258 | −100 | −110 | −110 | −107 | 365 | 72 | 73 | 8 | 8 | 7 | 7.7 | 2.87 | 4.53 | 3.55 | 3.65 |
| Adamo | 266 | −100 | −100 | −100 | −100 | 1 | 72 | 74 | 11 | 7 | 7 | 8.3 | 3.32 | 2.74 | 2.65 | 2.90 |
| **Bergamotto** | 224 | −100 | −100 | −110 | −103 | 327 | 112 | 75 | 11 | 7 | 7 | 8.3 | 3.32 | 4.45 | 5.76 | 4.51 |
| S. Caterina | 201 | −100 | −100 | −100 | −100 | 301 | 138 | 75 | 11 | 7 | 7 | 8.3 | 3.36 | 2.74 | 2.65 | 2.92 |
| **Spineddu** | 298 | −100 | −100 | −100 | −100 | 33 | 42 | 76 | 11 | 7 | 7 | 8.3 | 3.45 | 5.34 | 4.87 | 4.55 |
| **Ucciardone** | 298 | −100 | −100 | −100 | −100 | 33 | 42 | 76 | 11 | 7 | 7 | 8.3 | 3.89 | 4.82 | 4.64 | 4.55 |
| Cavaliere | 248 | −100 | −100 | −110 | −103 | 351 | 90 | 77 | 11 | 7 | 7 | 8.3 | 4.69 | 5.08 | 3.97 | 4.54 |
| Faccia Bedda | 206 | −100 | −100 | −100 | −100 | 2 | 74 | 77 | 11 | 7 | 7 | 8.3 | 3.00 | 3.33 | 4.77 | 3.70 |
| Muscatello | 196 | −100 | −100 | −100 | −100 | 296 | 145 | 77 | 11 | 7 | 9 | 9.0 | 3.89 | 5.87 | 3.94 | 4.57 |
| **S. Giovanni** | 165 | −100 | −100 | −100 | −100 | 265 | 176 | 77 | 9 | 11 | 9 | 9.7 | 4.62 | 3.24 | 2.98 | 3.61 |
| **Bianchetto** | 227 | −100 | −100 | −110 | −103 | 330 | 112 | 78 | 11 | 7 | 7 | 8.3 | 3.36 | 4.55 | 5.76 | 4.56 |
| Iazzolo | 227 | −100 | −100 | −110 | −103 | 330 | 113 | 79 | 7 | 7 | 7 | 7.0 | 2.65 | 2.72 | 2.63 | 2.66 |
| **Putiro d'Inverno** | 298 | −100 | −100 | −110 | −103 | 36 | 42 | 79 | 7 | 7 | 7 | 7.0 | 4.55 | 4.33 | 4.76 | 4.55 |
| S. Pietro | 181 | −100 | −100 | −110 | −103 | 284 | 159 | 79 | 7 | 7 | 7 | 7.0 | 2.65 | 2.72 | 2.63 | 2.66 |
| **Zio pietro** | 196 | −100 | −100 | −120 | −107 | 303 | 140 | 79 | 11 | 2 | 2 | 5.0 | 4.58 | 3.33 | 5.76 | 4.56 |
| **Urzì** | 278 | −100 | −100 | −100 | −100 | 13 | 68 | 82 | 7 | 7 | 7 | 7.0 | 4.92 | 3.22 | 5.87 | 4.67 |
| Butirra | 191 | −100 | −100 | −100 | −100 | 291 | 156 | 83 | 9 | 11 | 9 | 9.7 | 3.02 | 3.24 | 3.00 | 3.09 |
| Faccia donna | 206 | −100 | −100 | −100 | −100 | 306 | 143 | 85 | 2 | 7 | 7 | 5.3 | 1.76 | 2.55 | 2.11 | 2.14 |
| **Coscia** | 206 | −100 | −100 | −110 | −103 | 309 | 143 | 88 | 11 | 7 | 7 | 8.3 | 4.87 | 5.43 | 3.20 | 4.50 |
| **Campana** | 182 | −100 | −100 | −110 | −103 | 285 | 169 | 90 | 12 | 7 | 7 | 8.7 | 3.48 | 2.68 | 4.66 | 3.61 |
| **Reale** | 283 | −100 | −100 | −110 | −103 | 21 | 68 | 90 | 7 | 7 | 7 | 7.0 | 2.65 | 2.72 | 2.59 | 3.65 |
| **Regina** | 248 | −110 | −100 | −110 | −107 | 355 | 99 | 90 | 12 | 7 | 7 | 8.7 | 4.02 | 3.75 | 3.22 | 3.66 |

**Table 3.** Main phenological stages employed in the Chill Day model for the pears cultivated at the Experimental field 2 (850 m a.s.l.). Rows were ordered according to the bud sprouting data. For each accession, harvest date, Chill Days, chilling requirement, antichill days, bud sprouting date, threshold temperatures, root mean square error are reported. Chill days, threshold temperatures and root mean square error are presented both for the three years tested and as average values. Harvest date refers to the previous year compared to the sprouting bud. Accessions in bold were planted in both experimental fields. Chilling requirement and bud sprouting are expressed as doy (day of the year).

| Cultivar | Harvest-Day of the Year | Chill Days (Cd) [No. of Days] | | | | Chilling Requirement ($C_R$) [doy] | Anti-Chill Days (Ca) [No. Days] | Bud Sprouting [doy] | Threshold Temperature ($T_C$) (°C) | | | | Root Mean Square Error ($R_{MSE}$) | | | |
|---|---|---|---|---|---|---|---|---|---|---|---|---|---|---|---|---|
| | | 2014–2015 | 2015–2016 | 2016–2017 | Average | | | | 2014–2015 | 2015–2016 | 2016–2017 | Average | 2014–2015 | 2015–2016 | 2016–2017 | Average |
| Zuccareddu | 247 | −149 | −100 | −100 | −116 | 116 | 78 | 76 | 6 | 5 | 5 | 5.5 | 2.28 | 2.14 | 3.40 | 2.61 |
| Fracconello | 216 | −150 | −100 | −100 | −117 | 117 | 109 | 77 | 5 | 8 | 5 | 6.1 | 2.28 | 2.14 | 3.00 | 2.48 |
| **Gentile** | 195 | −170 | −100 | −110 | −127 | 127 | 120 | 77 | 6 | 5 | 5 | 5.5 | 2.45 | 2.10 | 2.90 | 2.48 |
| **Reale** | 297 | −180 | −100 | −100 | −127 | 127 | 18 | 77 | 6 | 4 | 8 | 6.0 | 2.41 | 2.12 | 4.21 | 2.91 |
| San Cono | 195 | −150 | −100 | −100 | −117 | 117 | 130 | 77 | 6 | 6 | 7 | 6.2 | 2.28 | 2.14 | 4.33 | 2.92 |
| Farcuneddu | 155 | −150 | −100 | −100 | −117 | 117 | 141 | 79 | 5 | 4 | 10 | 6.3 | 2.28 | 2.12 | 3.03 | 2.48 |
| San Pauluzzo | 175 | −170 | −100 | −110 | −127 | 127 | 142 | 79 | 7 | 4 | 9 | 6.7 | 2.37 | 2.12 | 2.56 | 2.35 |
| Alisio | 277 | −150 | −100 | −110 | −120 | 120 | 48 | 80 | 5 | 7 | 9 | 7.0 | 2.28 | 2.55 | 2.60 | 2.48 |
| Bella di Giugno | 155 | −150 | −100 | −110 | −120 | 120 | 170 | 80 | 5 | 4 | 9 | 6.0 | 2.28 | 2.38 | 4.22 | 2.96 |
| Bianchettone | 175 | −150 | −100 | −110 | −120 | 151 | 119 | 80 | 5 | 4 | 9 | 6.0 | 2.28 | 2.12 | 3.15 | 2.52 |
| **Cavaliere** | 262 | −170 | −100 | −100 | −123 | 123 | 60 | 80 | 8 | 11 | 10 | 10.1 | 2.45 | 3.36 | 2.94 | 2.92 |
| **Faccia Bedda** | 221 | −170 | −110 | −100 | −127 | 127 | 97 | 80 | 8 | 4 | 10 | 5.5 | 2.45 | 2.14 | 3.35 | 2.65 |
| Iazzuleddu | 257 | −150 | −100 | −100 | −117 | 117 | 71 | 80 | 5 | 4 | 2 | 5.4 | 2.28 | 2.59 | 3.90 | 2.92 |
| Piccola dolce | 165 | −110 | −100 | −130 | −113 | 113 | 167 | 80 | 5 | 5 | 6 | 5.3 | 2.10 | 2.19 | 3.15 | 2.48 |
| San Giovannino | 175 | −170 | −100 | −110 | −127 | 127 | 143 | 80 | 7 | 4 | 9 | 6.7 | 2.37 | 3.37 | 2.97 | 2.90 |
| **Ucciardone** | 313 | −170 | −100 | −100 | −123 | 123 | 9 | 80 | 8 | 7 | 7 | 7.2 | 2.45 | 3.39 | 1.70 | 2.51 |
| **Bianchetto** | 180 | −150 | −100 | −100 | −117 | 117 | 149 | 81 | 11 | 11 | 10 | 10.5 | 2.45 | 3.39 | 1.72 | 2.52 |
| **Campana** | 195 | −170 | −110 | −100 | −127 | 127 | 125 | 81 | 5 | 5 | 6 | 5.3 | 2.00 | 1.33 | 2.58 | 1.97 |
| Pergolesi | 257 | −100 | −100 | −110 | −103 | 42 | 86 | 81 | 5 | 4 | 9 | 6.0 | 2.12 | 2.12 | 2.81 | 2.35 |
| Spineddu | 313 | −170 | −100 | −100 | −123 | 123 | 10 | 81 | 8 | 12 | 10 | 10.5 | 2.45 | 3.26 | 3.00 | 2.90 |
| Virgolese | 272 | −180 | −110 | −110 | −133 | 161 | 13 | 81 | 6 | 4 | 7 | 5.5 | 2.11 | 2.34 | 1.55 | 1.96 |
| Coscia | 221 | −170 | −100 | −100 | −123 | 123 | 103 | 82 | 9 | 11 | 10 | 10.0 | 2.11 | 2.34 | 2.42 | 2.29 |
| Duchessa d'Angiò | 205 | −140 | −100 | −100 | −113 | 113 | 129 | 82 | 5 | 4 | 9 | 6.0 | 2.28 | 2.14 | 3.13 | 2.52 |
| **Muscatello** | 210 | −170 | −100 | −100 | −123 | 92 | 145 | 82 | 9 | 11 | 11 | 10.5 | 2.41 | 3.54 | 3.18 | 3.04 |
| Paradiso o Confettaro | 205 | −150 | −100 | −110 | −120 | 120 | 122 | 82 | 5 | 4 | 9 | 6.0 | 2.28 | 2.10 | 3.18 | 2.52 |
| Rozzuolo rosato | 195 | −180 | −100 | −110 | −130 | 130 | 122 | 82 | 6 | 4 | 9 | 5.3 | 2.28 | 3.08 | 2.09 | 2.48 |

Table 3. *Cont.*

| Cultivar | Harvest-Day of the Year | Chill Days (Cd) [No. of Days] | | | | Chilling Requirement ($C_R$) [doy] | Anti-Chill Days (Ca) [No. Days] | Bud Sprouting [doy] | Threshold Temperature ($T_C$) (°C) | | | | Root Mean Square Error ($R_{MSE}$) | | | |
|---|---|---|---|---|---|---|---|---|---|---|---|---|---|---|---|---|
| | | 2014–2015 | 2015–2016 | 2016–2017 | Average | | | | 2014–2015 | 2015–2016 | 2016–2017 | Average | 2014–2015 | 2015–2016 | 2016–2017 | Average |
| **San Giovanni** | 210 | −100 | −100 | −119 | −106 | 87 | 150 | 82 | 7 | 7 | 6 | 6.7 | 2.12 | 2.55 | 2.90 | 2.52 |
| Spadona | 190 | −180 | −100 | −110 | −130 | 130 | 127 | 82 | 6 | 4 | 9 | 6.3 | 2.41 | 2.53 | 3.76 | 2.90 |
| Tabaccaru | 195 | −180 | −100 | −110 | −130 | 191 | 61 | 82 | 6 | 4 | 9 | 6.3 | 2.41 | 2.10 | 1.34 | 1.95 |
| **Urzì** | 292 | −140 | −100 | −110 | −117 | 117 | 38 | 82 | 6 | 7 | 6.4 | 6.5 | 2.65 | 3.22 | 1.45 | 2.44 |
| **Zio pietro** | 210 | −170 | −100 | −100 | −123 | 123 | 114 | 82 | 8 | 6 | 6 | 6.5 | 2.45 | 3.11 | 2.01 | 2.52 |
| **Bergamotto** | 243 | −150 | −100 | −100 | −117 | 101 | 104 | 83 | 9 | 10 | 9 | 9.5 | 2.28 | 3.41 | 1.90 | 2.53 |
| Bruttu Beddu | 257 | −140 | −110 | −100 | −117 | 117 | 74 | 83 | 5 | 10 | 2 | 5.7 | 2.28 | 3.08 | 2.23 | 2.53 |
| Buona Luisa | 277 | −180 | −100 | −110 | −130 | 130 | 41 | 83 | 6 | 4 | 9 | 6.3 | 2.28 | 2.10 | 3.15 | 2.51 |
| Chiuzza | 262 | −120 | −100 | −100 | −107 | 107 | 79 | 83 | 10 | 2 | 12 | 8.0 | 3.07 | 1.09 | 3.41 | 2.52 |
| Garibaldi | 205 | −140 | −110 | −100 | −117 | 117 | 126 | 83 | 5 | 10 | 2 | 5.7 | 2.28 | 3.08 | 2.63 | 2.66 |
| Garofalo | 247 | −180 | −100 | −110 | −130 | 130 | 71 | 83 | 6 | 4 | 9 | 6.3 | 2.41 | 2.40 | 4.00 | 2.94 |
| Moscatello Maiolino | 175 | −180 | −100 | −100 | −127 | 127 | 118 | 83 | 6 | 4 | 10 | 6.7 | 2.41 | 2.10 | 3.05 | 2.52 |
| Moscatello nero | 195 | −180 | −100 | −110 | −130 | 130 | 123 | 83 | 6 | 4 | 9 | 6.3 | 2.45 | 3.39 | 2.82 | 2.89 |
| **Pasqualino** | 308 | −140 | −100 | −100 | −113 | 113 | 27 | 83 | 9 | 11 | 10 | 10.1 | 2.45 | 3.36 | 1.74 | 2.52 |
| Piridda | 170 | −170 | −100 | −110 | −127 | 127 | 151 | 83 | 8 | 4 | 9 | 7.0 | 2.45 | 2.10 | 3.00 | 2.52 |
| Piru Pizzo | 200 | −170 | −100 | −110 | −127 | 127 | 121 | 83 | 8 | 4 | 9 | 7.0 | 2.41 | 2.10 | 1.50 | 1.97 |
| Pisciazzanu | 216 | −180 | −100 | −110 | −130 | 100 | 132 | 83 | 6 | 4 | 9 | 6.3 | 2.41 | 2.10 | 3.05 | 2.52 |
| Pistacchino | 262 | −180 | −100 | −110 | −130 | 113 | 73 | 83 | 6 | 4 | 9 | 6.3 | 2.41 | 2.10 | 3.48 | 2.66 |
| **Putiro d'Inverno** | 308 | −170 | −100 | −100 | −123 | 123 | 17 | 83 | 9 | 10 | 11 | 10.1 | 2.45 | 3.36 | 1.75 | 2.52 |
| Putiru d'estate | 195 | −180 | −100 | −110 | −130 | 130 | 123 | 83 | 6 | 4 | 7 | 5.7 | 2.41 | 2.10 | 3.04 | 2.52 |
| Razzuolo | 195 | −140 | −110 | −100 | −117 | 230 | 23 | 83 | 5 | 10 | 2 | 5.5 | 2.41 | 2.10 | 4.19 | 2.90 |
| **Regina** | 292 | −130 | −100 | −120 | −117 | 117 | 39 | 83 | 6 | 6 | 6 | 6.0 | 4.78 | 2.44 | 2.03 | 3.08 |
| Rosa | 247 | −180 | −100 | −110 | −130 | 403 | 71 | 83 | 6 | 4 | 8 | 6.0 | 2.71 | 2.35 | 3.22 | 2.66 |
| Savino | 195 | −180 | −100 | −110 | −130 | 161 | 95 | 83 | 6 | 4 | 9 | 6.3 | 2.41 | 2.10 | 2.53 | 2.35 |
| Catanese | 195 | −180 | −100 | −110 | −130 | 191 | 68 | 84 | 6 | 4 | 9 | 6.3 | 2.41 | 2.10 | 3.48 | 2.66 |
| Ialofaru | 205 | −140 | −100 | −100 | −113 | 113 | 132 | 85 | 5 | 6 | 6 | 5.7 | 2.28 | 2.12 | 1.43 | 1.94 |
| Ianculidda | 205 | −180 | −100 | −110 | −130 | 130 | 115 | 85 | 6 | 4 | 9 | 6.3 | 2.41 | 2.10 | 3.65 | 2.72 |
| Angelico doppio | 195 | −150 | −100 | −110 | −120 | 120 | 110 | 91 | 5 | 4 | 9 | 6.0 | 2.28 | 2.10 | 1.52 | 1.97 |

**Table 4.** Mean threshold temperatures and chilling requirements, registered in the experimental field 1 (10 m a.s.l) and experimental field 2 (850 m a.s.l.). For each parameter means were compared at pa 0.001 level (***), based on ANOVA.

| Experimental Field | Threshold Temperature [°C] | Chilling Requirement (Chill Days) [Days] |
|---|---|---|
| Experimental field 1 | 8.10 | −103 |
| Experimental field 2 | 6.70 | −120 |
| Sig. | *** | *** |

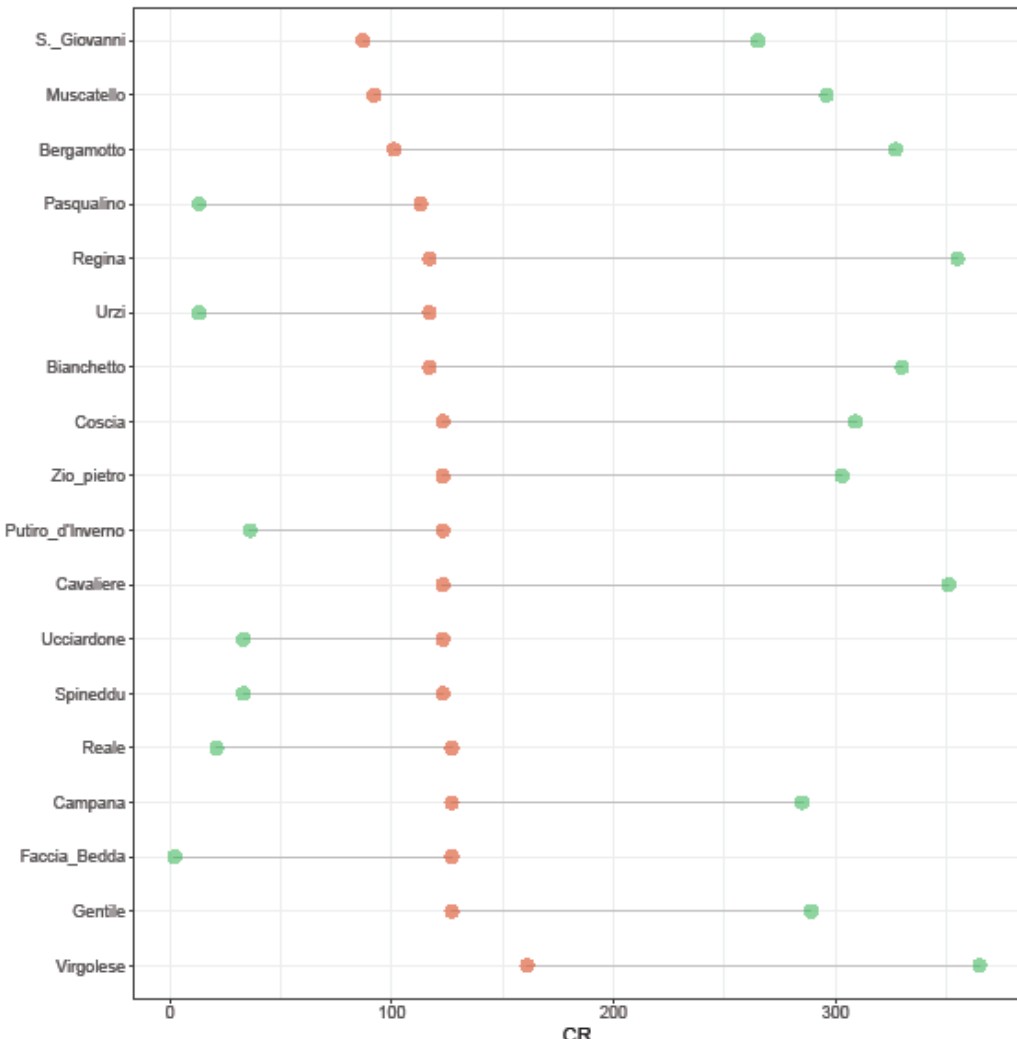

**Figure 3.** Chilling requirement (CR) of the 18 pear cultivars in both experimental fields (EF1 and EF 2). In green and red the CR values detected in EF1 and EF2, respectively.

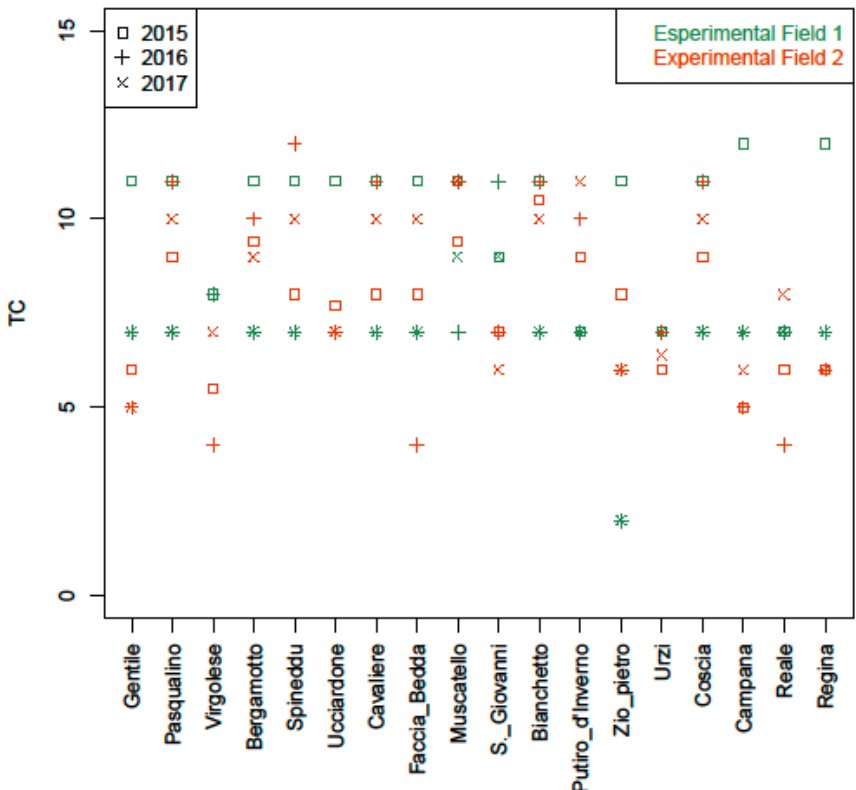

**Figure 4.** Threshold temperature (TC) expressed in Celsius degrees (°C) of the 18 pear cultivars in both experimental fields (EF1 and EF 2).

In EF1, TC was higher compared to that reported in previous reports for pear and cherry (6–7.5 °C), while results were similar to those detected for kiwi (8°), olive and some woody species (≥10 °C) [28,47]. On the other side, the mean CR values were very close to those reported by Cesaraccio et al. [28] for pear.

As for the phenological traits, the length of the quiescence period was strongly influenced by the high variability in harvest date among the studied cultivars [48] (Tables 2 and 3). In fact, harvest date began in the last days of June in both EFs and ended at the end of October in the EF1 and in the first week of November in the EF2. Therefore, in both EFs, the fulfilment of the CR for several early-ripening cultivars was reached in the third week of September or at the beginning of October while, for late-ripening cultivars, the CR was reached later in February (Tables 2 and 3). Among the 18 cultivars grown in both EFs, 'Reale', 'Campana', 'Regina' and 'Coscia' showed an earlier flower bud sprouting in the colder EF2 compared to EF1 (from 11 to three days). Anticipation of the flower bud sprouting in colder environments could be related to an easier fulfilment of the CR as already reported for 'Coscia' (Figure 3) [28,48]. The remaining fourteen cultivars showed flower bud sprouting anticipation in the warmer EF1 (six days on average), with 'Pasqualino' and 'Virgolese' showing the highest precocity (11 days, Figure 3). As for TC, no major differences were detected among the two experimental fields (Figure 4).

Figure 4 reports the registered TC for each season. As far as the EF1, several cultivars showed a strong variability between the first (>10 °C) and the third season (7 °C). In the EF2, for several cultivars the TCs registered in the first season were lower than those observed for EF1.

As reported in the literature, 'Coscia' is considered a low chilling requirement cultivar and, consequently, it is widely used in different warm winter countries. In our condition, the bud break for this cultivar occurred between the second and the third week of March, 14 days later to that observed in two different areas in Sardinia (Italy), in which the bud break was registered between the end of February and the first week of March [28],

but earlier compared to Lleida (Spain) in which bud break was registered at the end of March [49]. 'Spadona' and 'Coscia' are the main pear cultivars grown in the warm climate of Israel, and several agronomical and genetic studies have been carried out to establish the field performance and the genetic control of their bud-break time [50,51]. A cross population obtained by crossing 'Spadona' (low CU-requiring cultivar = 300 CU) and 'Harrow Sweet' (high CU-requiring cultivar = 800 CU) has been used for QTL fine-mapping of vegetative bud break time in European pear [46]. Our results highlighted that more than 80% of the accessions in EF1 showed an evident anticipation of bud break compared to 'Coscia'. Most of these cultivars showed an anticipation ranging from 16 to six days in bud break in EF1 and EF2 respectively, indicating the presence in this germplasm of interesting variability sources for this specific trait.

Table 5 reports the rates of inflorescence emergence in excided twigs of one year after an endodormancy period. After 14 days of observation only the cultivar 'Gentile' exceeded the 70% of sprouting for buds stored for 900 h at $4 \pm 0.1$ °C. The threshold of 70% of bud break is often considered the threshold to define the period of endo-dormancy breaking [11]. After 21 day at warm temperature (25 °C) the highest values of bud sprout were observed on twigs stored for 700 h, while the 900 h did not show the same effectiveness. Among the accessions in analysis, 'Bianchetto', 'Gentile' and 'Muscatello' exceeded 70% of bud sprout under the 700 h cold treatment. 'Gentile' exceeded the bud sprout threshold under 500 and 900 h of cold treatment as well, while bud sprout in 'Muscatello' occurred only at 900 h treatment. After 28 days at 25 °C, all accessions reached the 70% of bud sprout after 700 h. These results indicated that for the majority of the studied accessions, a cold exposition of at least 700 h, followed by a 21-day exposure to warmer temperature (25 °C in our conditions), are mandatory for break endodormancy and for sprouting. For all the accessions, the CU requirement was fulfilled after 700–900 h of cold exposition, except for 'Gentile' for which a lower chilling requirement between 500 and 700 h was evidenced. The low chilling requirements of 'Gentile' makes this cultivar an ideal candidate for cultivation in warm climates. The overall results evidenced a general low CU requirement of the analyzed accessions, compared to those reported for other varieties, indicating a CU requirement ranging from 1600 to 1800 hours [39]. In addition to the temperature effect, the two EFs were characterized by a different type of soil. EF2 in particular is characterized by a volcanic ash soil, thus, even though temperatures were lower, and rainfall was higher, plants were more stressed in EF2 compared to EF1.

**Table 5.** Inflorescence emergence rates observed in excised twigs of six early genotypes cultivated in the experimental field 2 (850 m a.s.l.). Excised twigs (one year old) were stored in climatic chamber for 300, 500, 700 and 900 h at +4 ± 0.1 °C followed by 7 (T7), 14 (T14), 21 (T21) and 28 (T28) days at +25 ± 0.1 °C. Degrees of freedom for the error term (DFE), mean square error (MSE), Fisher's F-test (F), and significance level (Sig.) from one-way ANOVA within cultivars separately for days of storage in climatic chamber at +25 ± 0.1 °C are reported. Inflorescence emergence rates exceeding the 70% (end of endodormancy) are underlined.

| Cultivar | Days | Hours | | | | DFE | MSE | F | *p* Value |
|---|---|---|---|---|---|---|---|---|---|
| | | 300 | 500 | 700 | 900 | | | | |
| Bianchetto | T7 | 0.0 | 0.0 | 0.0 | 0.0 | - | - | - | - |
| | T14 | 0.0 | 25.3 | 30.6 | 43.6 | 36 | 442.12 | 7.557 | <0.001 |
| | T21 | 16.4 | 44.3 | <u>72.1</u> | 60.2 | 36 | 562.04 | 10.419 | <0.001 |
| | T28 | 53.4 | 55.2 | <u>100.0</u> | <u>96.7</u> | 36 | 365.39 | 17.727 | - |
| Coscia | T7 | 0.0 | 0.0 | 0.0 | 0.0 | - | - | - | - |
| | T14 | 0.0 | 0.0 | 0.0 | 0.0 | - | - | - | - |
| | T21 | 0.0 | 0.0 | 0.0 | 0.0 | - | - | - | - |
| | T28 | 0.0 | 0.0 | <u>100.0</u> | 0.0 | 36 | $1.402 \times 10^{-28}$ | $1.78 \times 10^{32}$ | <0.001 |
| Gentile | T7 | 0.0 | 0.0 | 0.0 | 37.0 | 36 | 348.69 | 9.76 | <0.001 |
| | T14 | 0.0 | 31.0 | 62.1 | <u>73.9</u> | 36 | 217.40 | 50.73 | <0.001 |
| | T21 | 29.3 | <u>80.9</u> | <u>82.1</u> | <u>88.0</u> | 36 | 266.73 | 25.93 | <0.001 |
| | T28 | <u>85.4</u> | <u>80.9</u> | <u>96.1</u> | <u>95.0</u> | 36 | 237.04 | 2.29 | 0.095 |
| Muscatello | T7 | 0.0 | 11.2 | 0.0 | 5.1 | 36 | 72.106 | 3.86 | 0.017 |
| | T14 | 0.0 | 0.0 | 7.5 | 66.2 | 36 | 172.81 | 72.42 | <0.001 |
| | T21 | 0.0 | 13.7 | 7.5 | <u>72.8</u> | 36 | 227.77 | 48.97 | <0.001 |
| | T28 | 12.4 | 13.7 | <u>90.0</u> | 62.8 | 36 | 198.272 | 96.76 | <0.001 |
| Ucciardone | T7 | 7.3 | 0.0 | 0.0 | 0.0 | 36 | 43.06 | 3.18 | 0.036 |
| | T14 | 17.3 | 11.7 | 10.3 | 16.4 | 36 | 300.04 | 0.41 | 0.748 |
| | T21 | 17.3 | 49.2 | 30.3 | 37.1 | 36 | 452.82 | 3.86 | 0.017 |
| | T28 | 31.3 | 49.2 | <u>100.0</u> | 62.1 | 36 | 905.05 | 9418 | <0.001 |
| Urzì | T7 | 0.0 | 0.0 | 0.0 | 0.0 | - | - | - | - |
| | T14 | 0.0 | 13.3 | 0.0 | 10.1 | 36 | 83.38 | 5.56 | 0.003 |
| | T21 | 0.0B | 13.2 | 21.0 | 10.1 | 36 | 126.46 | 5.91 | 0.002 |
| | T28 | 67.3 | 21.5 | <u>75.2</u> | <u>88.3</u> | 36 | 2030.46 | 4.16 | 0.013 |

## 4. Conclusions

The CD model adopted in this work showed its effectiveness in detecting site and genotype specific CR and TC. A precise determination of such parameters could allow a more precise identification of the cultivars that well adapt to specific environments, maximizing buds opening and ensuring an optimal development of the flower with positive influence in fruit yield and quality.

The studied accessions were selected by growers during the past centuries both for their good adaptability and fruit quality characteristics. Most of them showed a significant lower chilling requirement compared to that of the reference cultivar 'Coscia'. This trait is of particular interest for areas characterized by mild winters in which the adoption of low chilling varieties could allow the cultivation of temperate species, including pear, contributing to the enlargement of market availability thanks to a wide ripening calendar.

**Supplementary Materials:** The following are available online at https://www.mdpi.com/2311-7524/7/3/45/s1, Table S1: List of the studied genotypes during the period 2014–2017 grown in the two s experimental field 1 (10 m a.s.l) and experimental field 2 (850 m a.s.l.). Accessions in bold were planted in both experimental fields. Table S2: Daily minimum, mean and maximum air temperatures and rainfall registered in the experimental field 1 (10 m a.s.l) (A) and experimental field 2 (850 m a.s.l.). Climatic data were provided by the Sicilian Water Observatory (www.osservatorioacque.it (accessed on 20 April 2020)). Table S3: Results of the LSD pairwise comparison procedure at the p prot Significant Difference (LSD). Excised twigs (one year old) were stored in climatic chamber for

300, 500, 700 and 900 h at +4 ± 0.1 °C followed by 7 (T7), 14 (T14), 21 (T21) and 28 (T28) days at +25 ± 0.1 °C.

**Author Contributions:** F.F.: conception and design of the study, acquisition of data, analysis and interpretation of data, collection and assembly of data, drafting and revision of the article, final approval of the version to be submitted; D.G.M.: statistical expertise, analysis and interpretation of data, drafting and revision of the article, final approval of the version to be submitted; M.A.: statistical expertise, acquisition of data, analysis and interpretation of data, revision of the article, final approval of the version to be submitted; E.N.: analysis and interpretation of data, revision of the article, final approval of the version to be submitted; A.C.: acquisition of data, revision of the article, final approval of the version to be submitted; S.L.M.: analysis and interpretation of data, revision of the article, final approval of the version to be submitted; A.G.: analysis and interpretation of data, revision of the article, final approval of the version to be submitted: G.D.: conception and design of the study, acquisition of data, analysis and interpretation of data, drafting and revision of the article, final approval of the version to be submitted. All authors have read and agreed to the published version of the manuscript.

**Funding:** This research received no external funding.

**Institutional Review Board Statement:** Not applicable.

**Informed Consent Statement:** Not applicable.

**Data Availability Statement:** Raw data can be accessed thourgh the following link: http://gofile.me/3aPZV/UldZpGPld (accessed on 6 February 2021).

**Conflicts of Interest:** The authors declare no conflict of interest.

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
