# Peer review of "Assessment of Chilling Requirement and Threshold Temperature of a Low Chill Pear (Pyrus communis L.) Germplasm in the Mediterranean Area"

_horticulturae, doi:10.3390/horticulturae7030045_

Round 1

Reviewer 1 Report

Dear authors,

With great interest I have read your article on chill requirement (CR) and threshold temperature (TC). In your introduction you make clear of which importance your study could be for future cultivation scenarios und climate change. In this context both parameters (along with chill hours) measured are of major importance.

Unfortunately, this importance is somewhat hidden in the tables, where you present 3-year means. I have commented in the PDF file on it. My main point would be that you revise your manuscript to show the details of these data in separated tables. I would suggest to put these data in graphs. Graphs are much easier for the reader to understand on first sight. Tables are very long and you have to search in them to find the cultivars under both environments (although formatted in bold). Even with your description in results & discussion it is not always easy to follow.

Your manuscript would be improved, if you would add figures for the cultivars cultivated on both experimental sites and to present the data on CR and TC in detail. At least I would do it for these cultivars.

Instead of using date (harvest date, bud sprouting) I would use "day of the year" to make it easier for reader to compare values.

The first paragraph of your Results (better Results & Discussion) reads more like an introduction. You should shorten it.

Coming to your objectives:

(1) Phenology of 61 cultivars. You provide sufficient information. I would prefer also to have data for the single years. You could consider presenting these detailed data in supplementary files.

(2) Detect optimal CR and TC. Coming back to my statement made above. For this I strongly recommend to present data as detailed as possible along with means over the years. But to present only the means will take away some important information on variability over the years.

(3) Chill hours. Like comments made for (2). I would also present here the detailed data.

My impression is, that you should present the detailed data where you want to make your major claims. I think the reader expect to see the detailed data for this. And concerning climate change it would be interesting to see how much variability is within the data.

I have made more comments in the text. Four formal issues I would like to mention here: (1) there are a lot of unnecessary hyphen in the text, (2) most of the text is bold formatted, (3) you should give DOI numbers in the references, and (4) the tables need to be re-formatted (e.g. column headings span over rows, readability)

Statistics: You should apply a 2-way ANOVA (as commented in the text). If you have detailed data you could add statistical evaluation here too which would strengthen your manuscript. I think it is required for the three main topics: CR, TC, and chill hours.

Despite these comments, your manuscript is presenting a highly interesting study. It would be a pity, if this is only reflected in 3-year means. Your study would be a valuable reference for chilling requirements in low chill pears. You should take the time to revise the manuscript, because I think your data deserve it.

With that said I recommend the manuscript for "major revision" to make clear that I think that it would be important to revise the manuscript in a way I outlined above. But I think that can be done pretty fast with all data at hand.

Despite that I like very much that the article is as short as possible. I found it written in a concise manner. The introduction is well written. In results & discussion text needed to be adapted after changes in tables / graphs have been made. But important for me is, that it will not require a full revision. The data are already there and you have described them. Main issues are (1) the visualization and (2) detail level for the three parameters mentioned. With adding these details you might have to adapt the text.

Author Response

Dear authors,

With great interest I have read your article on chill requirement (CR) and threshold temperature (TC). In your introduction you make clear of which importance your study could be for future cultivation scenarios und climate change. In this context both parameters (along with chill hours) measured are of major importance.

RESPONSE: Dear reviewer, thanks for your comments. We take your suggestions into account in this new version of the manuscript leading in what we think is an improved version of the work

Unfortunately, this importance is somewhat hidden in the tables, where you present 3-year means. I have commented in the PDF file on it. My main point would be that you revise your manuscript to show the details of these data in separated tables. I would suggest to put these data in graphs. Graphs are much easier for the reader to understand on first sight. Tables are very long and you have to search in them to find the cultivars under both environments (although formatted in bold). Even with your description in results & discussion it is not always easy to follow.

Your manuscript would be improved, if you would add figures for the cultivars cultivated on both experimental sites and to present the data on CR and TC in detail. At least I would do it for these cultivars.

RESPONSE: Dear reviewer, thanks for the comment, we made significant changes in the text and figures. Two new figures are now included showing the differences in CR and Tc between the two experimental sites for the accessions grown in both fields. The yearly data, together with the relative means, are reported in tables 1,2 and 3. A new supplementary figure was included to show the daily temperatur (rather than summarized by months).

Instead of using date (harvest date, bud sprouting) I would use "day of the year" to make it easier for reader to compare values.

RESPONSE: Thanks for the comment, we changed the tables content accordingly.

The first paragraph of your Results (better Results & Discussion) reads more like an introduction. You should shorten it.

RESPONSE: Thanks for the comment, we shorten the text avoiding repetitions with material and methods.

Coming to your objectives:

(1) Phenology of 61 cultivars. You provide sufficient information. I would prefer also to have data for the single years. You could consider presenting these detailed data in supplementary files.

RESPONSE: Thanks for the comment, data are also presented by single years in main and supplementary tables.

(2) Detect optimal CR and TC. Coming back to my statement made above. For this I strongly recommend to present data as detailed as possible along with means over the years. But to present only the means will take away some important information on variability over the years.

(3) Chill hours. Like comments made for (2). I would also present here the detailed data.

RESPONSE: Thanks for the comment, 2 and 3, we changed the table accordingly

My impression is, that you should present the detailed data where you want to make your major claims. I think the reader expect to see the detailed data for this. And concerning climate change it would be interesting to see how much variability is within the data.

RESPONSE: We agree with your comment, we included more detailed information in the manuscript

I have made more comments in the text. Four formal issues I would like to mention here: (1) there are a lot of unnecessary hyphen in the text, (2) most of the text is bold formatted, (3) you should give DOI numbers in the references, and (4) the tables need to be re-formatted (e.g. column headings span over rows, readability)

RESPONSE: Thanks for the comment we adjusted the text accordingly

Despite these comments, your manuscript is presenting a highly interesting study. It would be a pity, if this is only reflected in 3-year means. Your study would be a valuable reference for chilling requirements in low chill pears. You should take the time to revise the manuscript, because I think your data deserve it.

With that said I recommend the manuscript for "major revision" to make clear that I think that it would be important to revise the manuscript in a way I outlined above. But I think that can be done pretty fast with all data at hand.

Despite that I like very much that the article is as short as possible. I found it written in a concise manner. The introduction is well written. In results & discussion text needed to be adapted after changes in tables / graphs have been made. But important for me is, that it will not require a full revision. The data are already there and you have described them. Main issues are (1) the visualization and (2) detail level for the three parameters mentioned. With adding these details you might have to adapt the text.

RESPONSE: We agree with the comment, the text has been extended to describe the novel figures and information provided in the tables. The comments on the PDF file has been tackled as well.

Reviewer 2 Report

This paper presents on the determination of min T and chilling requirement in 61 accessions from observations over three years at two locations. While this is not my specific area of expertise, I found the justification of the paper compelling, the Introduction instructive and the results sensible.

A comment on presentation: Fig 2  this seems to be results in the MandM section

li 226 The result of different Tc and CR for the two EFs was ascribed to'genetic background' and 'environment'.  This is key to understanding the significance of the results. Can explanation be added on the history of the two sites - are the accessions not from the same source trees? are the trees of the same age?  If the difference is die to 'environment'...in what way?  are the authors pointing to impact of factors other than temperature?  or that simple summative GDH is not an adequate description of temperature effect?

Minor comments

English needs some tweaking, eg li 48 what does 'one week anticipation' in bud break mean? li 226 'reconducted'?

Editing required throughout for hyphenated wordseg li 133 'de-tected' also li 134,165, 163, 296, 301

Intro: li51-53 what do these models calculate if not GDD?  expand description

Methods: li 100 give detail on how temperature was monitored

li107 GDH given for March 1-June 30.  What Tmin was used? was a Tmax used?

li 136 for clarity, state # of observations used in the analysisfor a given accession. Was it n=6?  (3 years by two locations)

Fig 2 how is a non-integer Tmin achieved from this data? what model was fit to the data?

general comment - when possible, avoid over-use of abbreviations to keep life easier for the reader

Author Response

This paper presents on the determination of min T and chilling requirement in 61 accessions from observations over three years at two locations. While this is not my specific area of expertise, I found the justification of the paper compelling, the Introduction instructive and the results sensible.

RESPONSE: Dear reviewer, thanks for your words of appreciation on the paper and the suggestions provided.

A comment on presentation: Fig 2  this seems to be results in the MandM section

RESPONSE: thanks for the comment, we decided to include this figure in the result section since the graphs was a results of the data recorded.

li 226 The result of different Tc and CR for the two EFs was ascribed to'genetic background' and 'environment'.  This is key to understanding the significance of the results. Can explanation be added on the history of the two sites - are the accessions not from the same source trees? are the trees of the same age?  If the difference is die to 'environment'...in what way?  are the authors pointing to impact of factors other than temperature?  or that simple summative GDH is not an adequate description of temperature effect?

RESPONSE: thanks for the comment, more information are now provided in the material and methods section. The differences among the 2 EFs were also due to the type of soil, the one in EF2 was volcanic has soil, so it was not ideal for cultivation. A new sentence has been included in the discussion section. 

 Minor comments

English needs some tweaking, eg li 48 what does 'one week anticipation' in bud break mean? li 226 'reconducted'?

RESPONE: thanks for the comment, the text has been changed to make these sentences clearer.

Editing required throughout for hyphenated wordseg li 133 'de-tected' also li 134,165, 163, 296, 301

RESPONE: thanks for the comment, we also noticed this problem, the text has been adjusted accordingly

Intro: li51-53 what do these models calculate if not GDD? 

RESPONE: indeed the models calculate the GDD; a new sentence has been added to make this aspect clearer.

Methods: li 100 give detail on how temperature was monitored

RESPONSE: Climatic data were provided by the Sicilian Water Observatory (www.osservatorioacque.it), we now added the information also in the material and method section.

li107 GDH given for March 1-June 30.  What Tmin was used? was a Tmax used?

RESPONSE: Calculation is based on the sum of the differences between the choice of a critical (midnight) and optimum (midday) temperatures during the period 1 march  (assumed as the data in which the chill satisfaction is reached and a threshold base temperature (10 °C)

li 136 for clarity, state # of observations used in the analysis 6for a given accession. Was it n=6?  (3 years by two locations). Fig 2 how is a non-integer Tmin achieved from this data? what model was fit to the data?

RESPONSE: we fit the Chill Day model described by Cesaraccio and colleagues. The process is based on the iteration temperature threshold (TC) and CR that best predicts the bud-burst dates. The iterated temperature intervals were set with a window of 0.1°C to make the C and CR estimate more precise. 

general comment - when possible, avoid over-use of abbreviations to keep life easier for the reader

RESPONE: thanks for the comment, we changed the text accordingly. 

Round 2

Reviewer 1 Report

Dear authors,

thank you very much for the revised form. The changes made improved the quality of the article. Especially the new figures 3 and 4 give a much better insight through visualization.

You have addressed every comment made adequatly and made substantial changes in the text.

In the process you should keep an eye on the formatting of tables. Table 3 for example is spanning over three pages. On each page the headlines of the table should be repeated. Please discuss with the editorial team which is the best way to format it.

I recommend to publish the Article.